# Individual Work Attitudes and Work Ability

**DOI:** 10.3390/ejihpe15040053

**Published:** 2025-04-03

**Authors:** Nicola Magnavita, Carlo Chiorri, Francesco Chirico, Igor Meraglia

**Affiliations:** 1Department of Life Sciences and Public Health, Università Cattolica del Sacro Cuore, 00168 Rome, Italy; francesco.chirico@unicatt.it (F.C.); igor.meraglia01@icatt.it (I.M.); 2Department of Educational Sciences, University of Genova, 16126 Genova, Italy; carlo.chiorri@unige.it; 3Health Service Department, Italian State Police, Ministry of the Interior, 00185 Milan, Italy

**Keywords:** work annoyance, work engagement, overcommitment, social capital, work ability index, psychosocial stress, capacity, productivity, health surveillance, health promotion

## Abstract

Work capacity depends on many factors, including the age and health status of the employee, but also on personal characteristics and attitudes, such as reduced tolerance of unfavorable working conditions (Work Annoyance, WA), excessive commitment to work (Overcommitment, OC), passion for work (Work Engagement, WE), and social interactions (Social Capital, SC). A total of 1309 workers who underwent a medical examination at work completed questionnaires on work attitudes and assessed their work ability using the Work Ability Score (WAS). The relationship between variables expressing work attitudes and WAS was studied using hierarchical linear regression and moderation analyses. WA is associated with low WAS values; SC is a positive predictor of WAS and moderates the effect of WA on WAS. OC reduces work ability, while Vigor and Dedication, components of WE, have a strong positive effect on work ability. To improve the work ability of employees, employers and managers should improve social relations in the workplace and discourage overcommitment. A positive working environment can increase engagement and avoid triggers of work annoyance.

## 1. Introduction

Job attitudes refer to evaluations of an individual’s job that reflect their feelings, beliefs, and attachment to it ([61]). These areas of research are among the oldest, most popular, and most influential in organizational psychology due to their predictive capabilities regarding job performance, job satisfaction, and mental health in the workplace ([60]). Research indicates that subjective perceptions of job characteristics serve as situational antecedents of job attitudes, and different models have been presented to explain how they affect psychological outcomes at work. Hackman and Oldham’s Job Characteristics Theory (JCT) identifies five core subjective job dimensions: skill variety, task identity, task significance, autonomy, and feedback ([44]). These dimensions predict critical psychological states, which subsequently predict personal and work outcomes.

Work ability is the ability of a worker to carry out the work assigned to him/her, and it depends on the balance between the demands of the job and the professional skills and physical-mental conditions of the worker. Because of how complicated the idea of work ability is and how different technologies and workers’ health conditions can be, it has been defined in different ways by different authors in the literature. It also seems to be a concept that changes over time rather than a static assessment ([54]; [147]; [71]). Several studies have found that people’s work skills get worse with age ([36]; [25], [26]; [65]; [40]; [148]; [129]), and this may be different for men and women ([63]; [102]; [37]; [124]). This is probably due to cultural and social factors. Other individual and work-related factors have shown a correlation with work ability (see, e.g., [154]). These include chronic medical conditions, such as cancer ([144]; [162]; [28]) or musculoskeletal disorders ([62]; [94]), but also psychosocial and lifestyle factors ([39]). The impact of organizational factors and individual behavior on work ability is well recognized ([29]; [38]). Poor work ability is a predictor of absenteeism ([111]) and is associated with increased cognitive failures ([1]). Not infrequently, poor work ability of workers can seriously interfere not only with the quality of production but also with the health and safety of users ([123]). The dynamic characteristic of work ability justifies efforts to understand the factors that can influence its level in order to develop the most effective interventions to improve it. Companies have conducted numerous interventions to promote the work ability of their employees ([104]; [110]; [114]). Hence, the interest in understanding whether workers’ work attitudes can influence work ability is increasing.

Numerous studies have investigated the relationship between workers’ attitudes and their work ability. There is no shared definition of the term “job attitudes”; consequently, different authors have given it different meanings. Historically, researchers’ interest has been primarily focused on the definition and measurement of work involvement ([74]) and on the relationship between work organization and job attitudes ([116]). The work involvement or emotional attachment people have toward their work is commonly measured by two variables, work engagement and overcommitment; the former has a positive value, while the latter has negative effects on health and well-being. Despite having a marginally positive relationship, overcommitment and work engagement are two distinct concepts ([137]; [138]). Although these two measures derive from the same set of resources, they have seldom been explored simultaneously. Work engagement is defined as a positive, fulfilling, work-related state of mind characterized by vigor, dedication, and absorption ([30]). Vigor refers to high levels of energy and mental resilience while working, the willingness to invest effort in one’s work, and persistence in the face of difficulties. Dedication comprises a sense of purpose, passion, motivation, pride, and challenge. Absorption is the state of being fully absorbed and enjoying one’s task, which makes time fly by and makes it difficult to break away from it.

An examination of the relationship between workers’ attitudes and work ability cannot ignore the involvement and interactions of the workforce, which are measured by variables such as social capital. Social capital refers to a multifaceted psychological construct that emphasizes the value of interpersonal relationships and the resources, advantages, and opportunities that people can access through their social networks, relationships, and social interactions at work. It can be thought of as a dynamic psychological resource that, through reciprocity, trust, shared standards, and social networks, promotes both individual and community results. This term refers to the resources a person may be able to access through their social connections. These resources are determined by the structure and pattern of social network links (structural dimension), the state of interpersonal connections, which includes emotional ties, trust, and respect (relational dimension), and the shared understanding, interpretations, and meanings among social groups (cognitive dimension) ([34]; [108]).

More recently, a new type of attitude toward work has come under study: preliminary intolerance toward common work demands, or “Work Annoyance”. Work annoyance can be defined as the level of irritation or frustration experienced by workers regarding specific aspects of their job. It pertains to the working conditions, namely, factors such as the physical work environment, safety, and available resources that can influence a worker’s comfort and efficiency, and to the cognitive demands of the work, namely, the mental requirements of the job, including workload, complexity of tasks, and the need for continuous learning ([82]). Studies demonstrated that work engagement and work annoyance have different antecedents and consequences ([64]).

We are unaware of any studies that have used these different measures simultaneously to verify their effect on work ability, but some research leads us to believe that each of them may be correlated with work ability.

The relationship between work attitudes and work ability can be explained by several psychological mechanisms: affective mechanisms, such as job satisfaction and motivation or emotional regulation, help workers maintain high levels of motivation and to cope with stress, thus preserving work ability ([61]; [77]). Cognitive mechanisms, such as self-efficacy, perceived control, and job crafting, are important for work ability since they make workers feel in control of their work environment and are capable and actively modifying their tasks to align with their strengths ([154]; [150]). Work engagement, which comprises cognitive, emotional and physical engagement ([51]), enhances work ability in nursing professionals ([151]). In longitudinal studies ([95]; [46]), overcommitment, which is often used interchangeably with workaholism or work addiction to describe a worker’s tendency to be too committed to his or her own work ([9]), has been found to be a strong predictor of poor work ability in healthcare workers. Organizational social capital also proved to be associated with work ability ([66]). In healthcare workers, work annoyance (the subjective assessment of work characteristics, such as working conditions and cognitive demands, as something that bothers them) is negatively linked to work ability ([89]). Furthermore, there exists a correlation between several attitudes. For example, in religious people, belief in the organization’s ability to respond to change is significantly correlated with work engagement and job satisfaction ([33]). We did not include job satisfaction among attitudes, as some researchers do, because we believe that satisfaction derived from work (such as well-being, self-esteem, happiness, and self-efficacy) is a consequence and not a determining factor ([2]; [121]; [103]).

Although the studies reported above provide indications of the relationships between some attitudes and work ability, to the best of our knowledge, no study has simultaneously investigated the pattern of association of such attitudes and their relative impact on work ability. Therefore, we addressed this issue by collecting data on a heterogeneous sample of workers from different work sectors.

## 2. Materials and Methods

During the medical examination that all employees who are exposed to occupational risks must undergo by law, 1438 workers were asked to fill out a questionnaire containing personal data, as well as questionnaires on individual working attitudes and work ability. Workers belonged to 19 companies that were monitored by our occupational medicine unit in 2019 in Lazio, Italy, and worked in various production sectors (industry, commerce, health, social assistance, and offices).

The eligibility criteria were to be called for the periodic visit to the workplace and to agree to carry out the survey by signing the informed consent form. Those who did not wish to participate or did not complete the survey were excluded.

We followed the Declaration of Helsinki when conducting the study. By signing the personal health document, the participants gave their informed consent. In line with the International Commission on Occupational Health’s (ICOH) code of ethics ([53]) and occupational medicine confidentiality principles, they also agreed that their personal data could be analyzed and the results shared in a way that made them anonymous. Ethics approval was granted by the Catholic University Ethics Committee (ID 3008, 5 June 2020). Because the study was cross-sectional, missing data could not be imputed, and the results were based only on completed survey responses.

Work ability was assessed with the Work Ability Score (WAS), which is the first item of the Italian version ([21]) of the Work Ability Index ([152]), the most widely used questionnaire for assessing work ability. WAS measures current work ability compared to the best of the worker’s lifetime. This short measure has a strong correlation with the longer version of the questionnaire ([32]) and is widely used in surveys on large populations ([35]; [132]; [101]).

Work Annoyance (WA), which is the level of annoyance that workers feel about certain aspects of the job, has been measured with the Work Annoyance Scale ([82]), a self-reported measure consisting of nine items. On each item, participants rate how annoyed they are with certain parts of their job using an 11-point Likert scale, where 0 means “No irritation” and 10 means “Utmost irritation”. The items outline prevalent working conditions, such as night shifts and cognitive demands, including the necessity to acquire a new language, which are frequently cited by workers as sources of dissatisfaction. The 9 questions add up to a final score that can range from 0 to 90. The Cronbach alpha coefficient in this study was 0.797.

Workplace Social Capital was measured using the eight-item scale proposed by [69] ([69]), the Italian version ([20]). Each item is scored with a 5-point scale from 1 = “fully disagree” to 5 = “fully agree”. The final score ranges from 8 to 40. Reliability (Cronbach’s alpha) was 0.929.

The Italian version ([86]) of Siegrist’s short-form questionnaire ([141]) was used to measure overcommitment, which is the intrinsic part of the effort–reward model of stress ([140]). Six items were scored on a 4-point scale, with possible scores ranging from 6 to 24. Cronbach’s alpha was 0.748.

Work engagement has been measured with the Italian version ([10]) of the Utrecht Work Engagement Scale—UWES ([130]). The shortened version of the questionnaire contains nine items, with a score range of 0–6. The questionnaire contains three components, Vigor, Dedication, and Absorption, each referring to three items. Cronbach’s α of the components was 0.842 for Vigor, 0.899 for Dedication, and 0.719 for Absorption, while for the total score, including all nine items, it was 0.907.

### Statistical Analyses

The distribution of the scores obtained from the questionnaires was initially studied using mean, median, and standard deviation. We used the Kolmogorov–Smirnov and Shapiro–Wilk tests to determine the normality of the distribution of the variables and resorted to robust methods when this assumption was not met.

We used hierarchical linear regression models to test the predictive value of each individual work-related variable in terms of work ability. In Model I, we set Work Annoyance as the focal predictor, adjusting for age and sex. Subsequently, we introduced Social Capital as a further predictor (Model II). In Model III, we included overcommitment, and in the final model, we also included the components of work engagement (Model IV).

We also specified two moderation models. In these models, Work Annoyance (i.e., unfavorable working conditions) was the predictor, Social Capital was the moderator and work ability was the response variable. In Model V, we entered sex and age as covariates; in Model VI, we added overcommitment and work engagement components (Vigor, Dedication, and Absorption) as additional covariates.

Statistical analyses were carried out using IBM/SPSS Statistics for Windows, Version 28.0 (IBM Corp.: Armonk, NY, USA) and *R* statistical package software, Version 4.4.3.

## 3. Results

The study involved 1309 employees (491 men, 37.5%; 818 women, 62.5%) out of a total population, composed of all workers classified as “at professional risk” in companies, of 1512 people (participation rate 86.6%). Table 1 reports the characteristics of the sample. Kolmogorov–Smirnov and Shapiro–Wilk tests were statistically significant, but this was likely to be an effect of the large sample size, as the discrepancy between means and the absolute values of the medians was negligible, and skewness and kurtosis were less than 1, except for the work ability scores.

Given the results of the descriptive analyses, the correlations between the study variables were calculated using Pearson’s coefficient and the percentage bend correlation coefficient ([161]) as a robust method.

As shown in Table 2, the two methods of estimating correlations provided very similar results. We observed a significant correlation between all attitudes at work. Annoyance was higher in females (who also reported lower levels of Social Capital and higher levels of Overcommitment) and in older workers. Higher levels of Annoyance were associated with higher levels of Overcommitment and with lower levels of Social Capital, Work Engagement, and Work Ability. Higher levels of Social Capital were associated with higher levels of Work Engagement and Work Ability and lower levels of Overcommitment. Higher levels of Overcommitment were associated with lower levels of Work Engagement and Work Ability. Work Engagement and Work Ability were positively correlated. Age had a positive correlation with Overcommitment and an inverse relationship with Social Capital, Work Engagement, and Work Ability (Table 2).

We then specified hierarchical linear regression models and tested them with the usual ordinary least squares (OLS) method and the robust method implemented in the *rlm* function in the *MASS* package (version 7.3-65) ([158]) in *R*. In each model, we computed the Variance Inflation Factor (VIF) for each predictor to test for collinearity, and in this case, we used ridge regression to address this issue. Since the estimates of the regression coefficients obtained with the robust methods were not substantially different from those obtained with OLS, the latter was reported here only (Table 3). The estimates obtained with the robust methods are reported in the Appendix A, along with VIF values.

In Model I, we observed that Work Annoyance was negatively related to Work Ability and to age. In Model II, Work Annoyance was still a significant negative predictor of Work Ability, but Social Capital emerged as a significant positive predictor. In Model III, these effects were still significant, and Overcommitment was a significant negative predictor of Work Ability. In Model IV, the effect of Social Capital and Overcommitment was no longer statistically significant, while we observed a significant positive effect of Vigor and Dedication, along with a significant negative effect of Overcommitment and age.

We also found that Social Capital moderates the effect of Work Annoyance on Work Ability. The interaction between the two variables was significant both in the model adjusted for age and sex (Model V) and in the model that included all variables (Model VI) (Table 4). Figure 1 shows that if there are high levels of Social Capital, Work Ability increases as Work Annoyance increases, while for lower levels of Social Capital, Work Ability is unrelated or decreases as Work Annoyance increases.

Work health surveillance allowed us to collect other elements of interest in prevention, which were not reported in this study because they do not influence the relationship between work attitudes and work ability. For example, the performance of night work was associated with work ability but did not affect the results presented here. Night workers (n = 306, 23.4% of the sample) had a significantly lower Work Ability score than other workers (7.92 ± 1.80 vs. 8.27 ± 1.73, Robust *F*(1, 1278) = 12.550, *p* < 0.001, η^2^ = 0.007 [0.001; 0.019]), but by specifying this variable as an additional predictor in Model IV, the significant of the effects of overcommitment, Vigor, and Dedication did not change (see Appendix A). Similarly, the significance of the moderation effect of social capital 4 remained unchanged after adjusting Model VI for night work (see Appendix A).

## 4. Discussion

The aim of this study was to investigate the different contributions that work annoyance, social capital, overcommitment, and the components of work engagement can provide in predicting work ability since no previous study had considered these dimensions in the same model. We found that higher work ability was significantly predicted by lower levels of overcommitment and higher levels of two components of work engagement, vigor and dedication. Furthermore, while the main effect of social capital and work annoyance was not statistically significant, their interaction indicated a moderation effect of the former. When social capital is high, work ability is higher even in the presence of high levels of work annoyance, while when social capital is low, work ability tends to decrease with the increase of work annoyance. Taken together, these results supported the view that work attitudes have a significant association with work ability and that an individual’s ability to perform a job safely and obtain satisfaction and well-being from it depends on his/her attitudes.

The work ability of employees is an extremely important parameter for company productivity. Work ability is associated with absenteeism ([109]; [76]) and presenteeism ([135]; [43]; [52]). Many interventions have been conducted in the workplace to reduce absenteeism and increase work ability and productivity. A meta-analysis showed that improving work ability absenteeism can be substantially reduced ([146]); however, evidence of the effectiveness of interventions on work ability is still moderate ([19]; [156]). Among the factors that reduce work ability, physical factors are particularly relevant. As people age, sarcopenia and cognitive decline ([55]; [118]), as well as having long-term diseases that make it difficult to work, such as neoplasms ([13]; [14]; [142]; [85], [91]), post-Covid ([112]) or chronic inflammatory diseases ([164]; [41]; [23]), contribute to a decrease in work ability. Metabolic disorders, such as diabetes, obesity, dyslipidemia, and hypertension, are other known physical risk factors for poor work ability ([145]; [8]; [45]; [98]). However, reduced work ability may also be a consequence of workplace violence ([81]; [87]), but it is also true that workers with reduced work ability are more exposed to workplace violence than their colleagues ([88]). Work-related stress ([4]; [122]), sleep problems ([72]; [157]; [15]), and common mental disorders such as anxiety and depression ([90]; [100]) are also associated with poor work ability. Therefore, the combination of these findings in the literature indicates that work ability strongly depends on physical and environmental factors that are independent of the individual’s approach to work. However, as we have found in this work, the attitude that each person has towards their work also influences work ability.

The first characteristic that we have considered is Work Annoyance, which represents what the worker feels before being exposed to an unfavorable working condition. Reduced ability to tolerate critical working conditions is inversely correlated with work ability. The Work Annoyance Scale taps into several features of the job that employees frequently complain about and find unpleasant. Some of these have to do with working conditions, such as long hours, working at night, commuting to work, physically demanding jobs, or stressful work environments. Other problems have to do with the cognitive demands of the job, such as learning how to use new electronic equipment, learning a foreign language, putting in a lot of effort to solve a problem at work, or learning new working methods. Workers are asked to report how much exposure would cause them discomfort if they were exposed to it. Therefore, this is an attitude that precedes work-related stress and is based on personal characteristics and previous work experience rather than the current situation.

In this study, the level of annoyance increased with age and was higher in females than in males. The degree of annoyance depends on the skills and training of the employee, as well as on previous work experience and placement in the current job. Certainly, a worker who has a particular intolerance to certain aspects of the job will react even more strongly if that task is frequently required of them. All these elements have been found to be associated with an increase in mental health problems (distress, anxiety, depression) or a decrease in employee well-being (satisfaction, happiness, and work engagement) ([99]; [12]). It may also play a key role in predicting psychological well-being (or lack thereof) ([82]). In this study, we observed that it also predicts poor work ability, and this suggests that its effect should be offset to improve the worker’s integration into the production activity. In the organization of work, intolerance towards some occupational requests can be attenuated or resolved through an assignment of tasks that respects the needs of each worker. Our study demonstrated that improving social capital can counteract the effect of work annoyance on work ability.

Social capital, which can be seen as a combination of social support, social networking, and social cohesion ([31]), can significantly moderate the relationship between Annoyance and Work Ability. It derives partly from the sociability characteristics of each worker, partly from the leadership style, and partly from the company’s working climate. There are two overarching conceptualizations of social capital, namely individual social capital and collective social capital ([22]). Social capital is all the potential and actual resources of a person, both material (such as titles, operational power, and organizational level) and immaterial (such as education, social standing, and political power). It comes from having a network of stable, reliable, and mutually beneficial social ties. In other words, social capital encompasses (a) the resources acquired and embedded within the social network’s trade process. The actual social network serves as an additional resource. Workers use these tools to achieve goals, satisfy desires, or improve their ability to handle challenging circumstances ([73]; [17]).

Studies have shown that social capital plays a crucial role in the ability of populations to resist disasters ([153]). In the workplace, it can help workers to withstand production crises and downsizing, but also to react to the daily difficulties of work. In a calm and friendly work environment, in which the relationships between management and operators are regular and profitable, the positive effect of high levels of social capital can counteract the negative effect of work annoyance, and the resulting work capacity of workers can be high. Positive working relationships and individual resilience are important to individual well-being. In workplaces, social capital is associated with job performance and job satisfaction ([113]). High social capital has been associated with greater resilience and reduced psychosocial stress in linguistic and cultural minorities ([11]) and improved business performance in farmers ([58]). Analysis of data from the sixth European Working Conditions Survey, including 28,900 employees in 35 European countries, showed that social capital is associated with a lower risk of depression ([67]). Furthermore, longitudinal studies showed that social capital increases competence and prevents occupational burnout ([56]) and is associated with higher rates of return to work after chronic disease ([128]).

Social capital exerts its effects on the workplace through various mechanisms: making information available to workers, promoting the application of social norms, improving the action of health surveillance services, and offering psychosocial support networks ([131]). However, the most crucial development is whether the quality of social capital allows management to receive and accept workers’ opinions, provided that they are reasonable. The positive effect of social capital on work annoyance comes from the fact that a profitable network of relationships between workers and management allows the former to express their needs and the latter to insert them into efficient organizational planning. Management of occupational risks is more efficient if workers are involved in the risk assessment, surveillance, information, and audit process ([78], [79]). For this reason, in our occupational medicine activity, we have given greater importance to participatory ergonomics groups ([80]) to improve this through social capital. This also means accepting and building a policy of accepting constructive dissent ([6]). Employers and managers should work to improve social relationships in the workplace and increase the strength of the working group, making them less afraid to deal with operationally critical issues. Different leadership styles (destructive, supportive, or relationally focused) can significantly influence social capital and relationships in the workplace ([107]; [163]). Companies should pay attention to strategies that improve social interaction and increase levels of support, reciprocity, participation, and group reliance between workers. Workplace health promotion interventions include the enhancement of social capital as an essential key to success ([7]).

The commitment and motivation that each worker has with his/her work are crucial in defining work capacity. Work behaviors should be conceptualized by placing them on a continuum from withdrawal/under-engagement (e.g., persistent absenteeism) to over-engagement or overcommitment (e.g., work conflicting with all other activity). Our study highlights that there is a clear difference between proper motivational commitment and work engagement and an excess of these. Overcommitment is a dimension describing an individual’s over-involvement toward their own work and an excessive effort and attachment to their job, with an incapability to detach from it and negative consequences in the form of poor health ([9]). It can be characterized as a genuine addiction ([42]) and, as such, must be prevented ([24]), promptly identified ([120]) and treated ([155]).

Heavy work investment, a broad concept that encompasses all types of self-investment in the workplace, can have both positive and negative aspects for workers. If the vigorous dedication of one’s resources for the work task has a positive effect on work ability, overcoming one’s limits is certainly negative. When workers strive to excel at work, disregarding individual conditions, past work experiences, family obligations, and personal life, they can negatively impact their work ability. Therefore, the worker should be passionate about his or her work but not excessively involved in the effort. According to a French study, overcommitment is related to the individual’s self-concept, reflecting a propensity to prioritize individual accomplishments and success. Over time, this is also associated with a much higher risk of developing depression. On the other hand, work engagement is strongly correlated with collective self-concept, which involves defining oneself through group membership. It protects workers against the risk of emotional fatigue and despair ([3]). Therefore, managers should encourage their employees to develop collaboration rather than competition.

In this study, overcommitment was associated with low work ability. This result of our study is in line with the literature. High overcommitment was associated with poor work ability in Brazilian nurses ([96]), low job satisfaction in doctors ([18]), psychological distress among university students ([115]), poor mental health, anxiety and depression in priests ([70]), and depressive symptoms, suicidal ideation, and risk of suicide in German veterinarians ([133]). Longitudinal analyses of data from the Third German Sociomedical Panel of Employees confirmed that overcommitment causes mental health problems in employees aged 40 to 54 years ([49]). Overcommitment has also been associated with distress, musculoskeletal, and mental disorders ([92]), pain ([126]), immune disorders ([105]), cardiovascular reactions ([140]; [149]), and impaired coronary microvascular function ([159]). Overcommitment is associated with vital exhaustion, which has been shown to predict the progression and manifestation of cardiovascular disease ([119]). Corporate social responsibility should lead to counteracting overcommitment due to its negative effects on public health. Our study shows that companies have a vested interest in counteracting this attitude because it reduces work ability. Work-family conflict and family-work conflict are significantly associated with overcommitment ([59]), and overcommitment exposes workers to greater job insecurity ([127]). Therefore, companies must avoid rewarding their employees’ excessive commitment and discourage the tendency to forgo rest and recovery because excessive fatigue is the cause of occupational pathology ([134]; [47]). Unfortunately, this line of thinking does not always guide business dynamics. In recent years, significant changes in businesses around the world, like the widespread adoption of remote working and digital transformation, have led to a blurring of the lines between work and personal life, raising questions about whether people’s investments in their jobs could negatively impact their lives and health. The style of managerial leadership plays a decisive role in promoting excessive commitment to work attitudes, which can end up resulting in stress, dissatisfaction, poor health among workers, and reduced productivity. For instance, it has been found that bosses who are too pushy and ask workers to put in extra hours of work can lead to workaholism and worsen workers’ mental health ([92]) and that poorly organized work is linked to both physical and mental health issues in workers ([83], [84], [91]).

However, this study indicated that work engagement has a very positive effect on work ability and exceeds overcommitment. Therefore, proper organization and effective management must develop engagement and avoid overcommitment. These factors that motivate people are more significant than work annoyance and social capital in the final hierarchical regression model, which shows that they are the most important in determining how well people can do their jobs. Vigor and dedication are the components of work engagement that play the greatest role in defining work capacity. Previous studies have shown that vigor and dedication protect workers from burnout ([48]). In particular, vigor has been shown to mediate the effect of different leadership styles on employee engagement ([75]). In our study, absorption does not have a significant relationship with workability. This is consistent with the results of a meta-analysis in which absorption has a smaller effect on turnover intention and job satisfaction than vigor and dedication ([97]).

The results of our survey, supported by data from the literature, provide clear indications of organizational psychology to companies. Overcommitment and work engagement can be seen as a continuum that reflects the faculties that an employee utilizes in their work. Management should strive to foster a strong passion for work, preventing it from becoming an obsessive pursuit of personal success at the expense of other aspects of life (overcommitment). This can be achieved by wisely using social capital, developing communication skills, and sharing resources, all hallmarks of work engagement. In our experience, overly engaged workers can appear extraordinarily productive, but an unexpected setback, such as a serious illness, can easily push them to the breaking point. For example, a manager at a large multinational company who was diagnosed with breast cancer chose to quit her job so as not to be a barrier to her colleagues ([85]).

The relationship between attitudes and workability that we observed was also expected in the literature. Perceived job characteristics associated with intrinsic motivation are associated with improved well-being and reduced burnout in nurses. Workers benefit from a positive work environment created by engaged leaders, which improves both their motivation and their well-being at work ([68]). Work engagement is characterized as the main antidote to burnout. Management strategies to promote it are relatively simple and inexpensive ([136]). Higher engagement in work was related to higher work ability in a Finnish sample of firefighters ([5]), in a cross-sectional study of aged workers ([160]; [143]), and in an observational study among physicians in Dutch hospitals ([27]).). A review of the literature has shown that in multinational companies, managing workers’ workability depends on management commitment and support, an effective communication strategy, and work engagement ([139]). An effective improvement in the work ability of the working population can therefore be achieved by improving the quality of the organization and the leadership style. There are prospects for the development of studies on these topics, as well as for the implementation of workability promotion programs.

Workplace health promotion programs have an overall positive but heterogeneous effect across the different approaches ([57]). Programs aimed at improving work ability have had mixed results ([125]; [117]; [106]). Several authors who have critically reviewed the literature have suggested that such programs should focus on work engagement ([16]; [50]; [93]). We join this invitation, which corresponds to what we observed in our study.

This study has several limitations. First, the sample, although large, was not randomized and, by definition, could not be considered representative of the entire Italian population of workers. However, it should be noted that it represented a very high percentage of the accessible population of workers who came for periodic visits to the occupational physician at work. The high participation in the proposed health promotion program suggests that selection bias was negligible. The study design, which is cross-sectional, does not allow inferences about the causality of the phenomena, even if the sequence proposed in the hierarchical regression is likely the most plausible. Work ability was measured with a single question, and although the item used was shown to correlate strongly with the complete WAI questionnaire, this nevertheless reduced the power of the statistical analyses. The choice of the four attitudes we measured excluded the possibility of assessing other attitudes potentially associated with work ability. Strengths include the recruitment of a fairly large number of workers from different production sectors who agreed to rate different types of attitudes toward work and to self-assess their own work ability. The results of the survey and suggestions to improve work ability were brought to the attention of employers and employees’ safety representatives during the periodic safety meetings that are mandatory in all companies in Italy.

## 5. Conclusions

Workers’ attitudes are significantly associated with work ability. Low tolerance to unfavorable working conditions (work annoyance) predicts low work ability, but proper management of social capital through careful enhancement of social relationships and constant support can effectively moderate the negative impact on work ability. The investment that each employee makes in their work must be appropriately cultivated from a collective perspective, allowing team spirit to prevail over individual success. In the propensities of each worker, work engagement must be cultivated with the right respect for recovery times and the needs of family life, thus excluding overcommitment. This emphasizes the importance of fostering a collaborative environment where teamwork is prioritized, as it can lead to improved overall performance and well-being among employees. By nurturing collective efforts, organizations can enhance individual and group work capabilities.

## Figures and Tables

**Figure 1 ejihpe-15-00053-f001:**
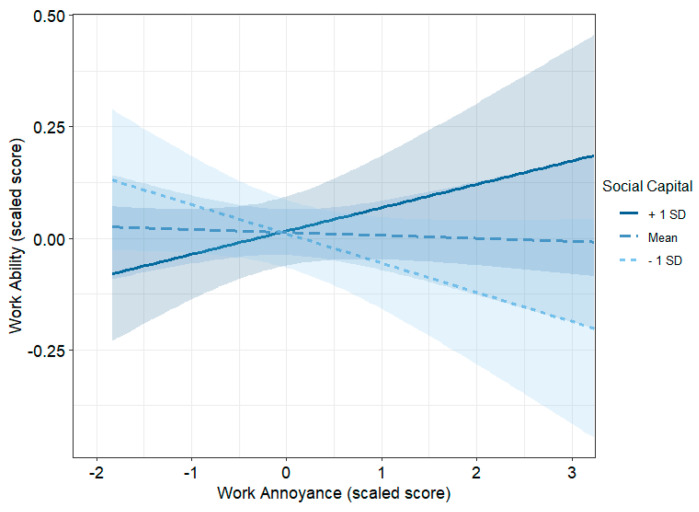
Interaction plot of the moderating effect of Social Capital in the interaction between Work Annoyance and Work Ability (from Model VI).

**Table 1 ejihpe-15-00053-t001:** Age, attitudes and work ability of the observed sample.

Variable	Range	Mean ± SD	95% CI	Median	Skewness	Kurtosis
Age	20–71	45.91 ± 10.7	45.33; 46.49	47	−0.21	−0.72
Work Annoyance	0–90	32.49 ± 17.81	31.52; 33.47	33	0.24	−0.22
Social Capital	8–40	26.21 ± 8.51	25.75; 26.67	26	−0.05	−0.88
Overcommitment	6–24	13.26 ± 3.51	13.07; 13.45	13	0.33	0.16
Work Engagement	0–54	37.08 ± 9.44	36.57; 37.59	37	−0.38	0.33
Vigor	0–18	11.92 ± 3.55	11.72; 12.11	12	−0.42	0.41
Dedication	0–18	12.84 ± 3.84	12.63; 13.05	13	−0.61	0.25
Absorption	0–18	12.33 ± 3.31	12.16; 12.51	12	−0.38	0.54
Work Ability	0–10	8.19 ± 1.75	8.09; 8.29	8	−1.37	2.76

Note. SD—standard deviation. 95% CI—confidence interval 95%.

**Table 2 ejihpe-15-00053-t002:** Bivariate correlation matrix between the variables used in this study. Pearson’s coefficients (lower triangle) and percentage bend correlation coefficients (upper triangle) with 95% confidence intervals.

	Female	Age	Annoy.	SocCap	Overcom.	Engag.	Work Ability
**Female**	1.000	0.024 [−0.032; 0.080]	0.134 *** [0.079; 0.188]	−0.087 ** [−0.142; −0.032]	0.058 * [0.002; 0.113]	−0.005 [−0.061; 0.051]	−0.038 [−0.093; 0.018]
**Age**	0.022 [−0.034; 0.078]	1.000	0.201 *** [0.147; 0.254]	−0.205 *** [−0.258; −0.151]	0.144 *** [0.089; 0.198]	−0.101 *** [−0.156; −0.046]	−0.133 *** [−0.187; −0.078]
**Annoy.**	0.137 ** [0.082; 0.191]	0.211 ** [0.157; 0.264]	1.000	−0.214 *** [−0.266; −0.160]	0.185 *** [0.131; 0.238]	−0.414 *** [−0.459; −0.367]	−0.227 *** [−0.279; −0.174]
**SocCap**	−0.080 ** [−0.135; −0.024]	−0.213 ** [−0.266; −0.159]	−0.230 ** [−0.282; −0.177]	1.000	−0.283 *** [−0.333; −0.231]	0.375 *** [0.326; 0.422]	0.201 *** [0.147; 0.254]
**Overcom.**	0.053 [−0.003; 0.108]	0.145 ** [0.090; 0.199]	0.195 ** [0.141; 0.248]	−0.288 ** [−0.338; −0.236]	1.000	−0.181 *** [−0.234; −0.127]	−0.199 *** [−0.252; −0.145]
**Engag.**	−0.007 [−0.063; 0.049]	−0.104 ** [−0.159; −0.049]	−0.421 ** [−0.466; −0.374]	0.388 ** [0.340; 0.434]	−0.184 ** [−0.237; −0.130]	1.000	0.403 *** [0.355; 0.449]
**Work** **Ability**	−0.027 [−0.083; 0.029]	−0.123 ** [−0.177; −0.068]	−0.198 ** [−0.251; −0.144]	0.202 ** [0.148; 0.255]	−0.212 ** [−0.265; −0.158]	0.400 ** [0.352; 0.446]	1.000

Note. *: *p* < 0.001; **: *p* < 0.01; ***: *p* < 0.05.

**Table 3 ejihpe-15-00053-t003:** Standardized regression coefficients with their 95% confidence interval and η^2^ measures of effect size with their 95% confidence interval of individual attitudes on Work Ability for four alternative hierarchical regression models.

	**Model I**		**Model II**	
**Variable**	**Beta**	**η^2^**	**Beta**	**η^2^**
Sex (Female)	0.013 [−0.041; 0.067]	<0.001 [0.000; 0.005]	0.019 [−0.035; 0.073]	<0.001 [0.000; 0.006]
Age	−0.078 ** [−0.133; −0.023]	0.006 [<0.001; 0.017]	−0.052 [−0.108; 0.003]	0.003 [0.000; 0.011]
Work Annoyance	−0.187 *** [−0.243; −0.131]	0.034 [0.017; 0.056]	−0.160 *** [−0.216; −0.103]	0.024 [0.010; 0.043]
Social Capital			0.149 *** [0.093; 0.205]	0.021 [0.008; 0.039]
	**Model III**		**Model IV**	
**Variable**	**Beta**	**η^2^**	**Beta**	**η^2^**
Sex (Female)	0.020 [−0.033; 0.074]	<0.001 [0.000; 0.006]	−0.005 [−0.056; 0.045]	<0.001 [0.000; 0.003]
Age	−0.043 [−0.098; 0.012]	0.002 [0.000; 0.010]	−0.053 * [−0.105; −0.001]	0.003 [0.000; 0.012]
Work Annoyance	−0.143 *** [−0.200; −0.087]	0.019 [0.007; 0.037]	−0.010 [−0.067; 0.048]	<0.001 [0.000; 0.004]
Social Capital	0.116 *** [0.058; 0.173]	0.012 [0.003; 0.027]	−0.003 [−0.059; 0.054]	<0.001 [0.000; 0.002]
Overcommitment	−0.132 *** [−0.188; −0.075]	0.016 [0.005; 0.033]	−0.090 ** [−0.146; −0.035]	0.008 [0.001; 0.021]
Vigor			0.234 *** [0.151; 0.317]	0.024 [0.010; 0.043]
Dedication			0.166 *** [0.085; 0.246]	0.013 [0.003; 0.028]
Absorption			0.020 [−0.050; 0.089]	<0.001 [0.000; 0.005]

*Note*. *: *p* < 0.001; **: *p* < 0.01; ***: *p* < 0.05.

**Table 4 ejihpe-15-00053-t004:** The moderating effect of Social Capital in the interaction between Work Annoyance and Work Ability.

	Model V		Model VI	
Variable	Beta	η^2^	Beta	η^2^
Sex (Female)	0.015 [−0.039; 0.070]	<0.001 [0.000; 0.005]	−0.009 [−0.060; 0.042]	<0.001 [0.000; 0.005]
Age	−0.055 [−0.111; 0.001]	0.014 [0.004; 0.029]	−0.055 * [−0.108; −0.003]	0.014 [0.004; 0.029]
Work Annoyance	−0.156 *** [−0.213; −0.100]	0.033 [0.016; 0.055]	−0.007 [−0.064; 0.051]	0.033 [0.016; 0.055]
Social Capital	0.155 *** [0.099; 0.211]	0.020 [0.008; 0.039]	0.003 [−0.054; 0.060]	0.020 [0.008; 0.039]
Work Annoyance × Social Capital	0.071 ** [0.019; 0.122]	0.005 [<0.001; 0.017]	0.059 * [0.011; 0.107]	0.004 [0.000; 0.014]
Overcommitment			−0.092 ** [−0.147; −0.036]	0.016 [0.005; 0.032]
Vigor			0.229 *** [0.146; 0.312]	0.094 [0.066; 0.126]
Dedication			0.165 *** [0.084; 0.246]	0.013 [0.004; 0.029]
Absorption			0.026 [−0.045; 0.096]	<0.001 [0.000; 0.005]

*Note*. *: *p* < 0.001; **: *p* < 0.01; ***: *p* < 0.05.

## Data Availability

Data are deposited on Zenodo 10.5281/zenodo.14869633.

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
