# Peer review of "Individual Work Attitudes and Work Ability"

_ejihpe, 2025, doi:10.3390/ejihpe15040053_

Round 1
Reviewer 1 Report
Comments and Suggestions for Authors
First, I would like to thank you for the opportunity to read this work. This study analysed variables expressing work attitudes and work ability using a sample of 1309 workers who underwent a medical examination at the workplace.
Below, I offer some comments and suggestions, hoping they will enhance the paper.
INTRODUCTION
As the authors added in the introduction, «Numerous studies have investigated the relationship between workers' attitudes and their work ability.». As such, it is important to highlight the novelty of this study and the gaps this study may help to fulfil. At the end of this section, the authors briefly highlighted the contribution of the study, but it is essential to develop the argumentation.
In addition, it is unclear why the authors chose the workers’ attitudes they studied instead of other attitudes. The literature on workers' attitudes needs to be deepened, and the literature on work ability must also be developed.
MATERIAL AND METHODS
It is unclear whether the instruments used were validated for the population from which the authors collected a sample.
RESULTS
Please indicate the country where the data was collected.
DISCUSSION
This section could be improved if the authors first discussed the results obtained and compared them with previous literature. Then, they could present the study's limitations and suggest avenues for future studies. Finally, they could finish this section by discussing the study's theoretical and practical implications.
Comments on the Quality of English LanguageThe English can be improved.
Author Response
First, I would like to thank you for the opportunity to read this work. This study analysed variables expressing work attitudes and work ability using a sample of 1309 workers who underwent a medical examination at the workplace.
Below, I offer some comments and suggestions, hoping they will enhance the paper.
Response: We sincerely thank the reviewer who made important contributions to improve the quality of the manuscript.
INTRODUCTION
As the authors added in the introduction, «Numerous studies have investigated the relationship between workers' attitudes and their work ability.». As such, it is important to highlight the novelty of this study and the gaps this study may help to fulfil. At the end of this section, the authors briefly highlighted the contribution of the study, but it is essential to develop the argumentation.
In addition, it is unclear why the authors chose the workers’ attitudes they studied instead of other attitudes. The literature on workers' attitudes needs to be deepened, and the literature on work ability must also be developed.
Response: We appreciated the reviewer's advice that gave us the opportunity to develop the Introduction, better explaining the necessity and originality of this work.
The reviewer rightly asked us to explain why we chose to investigate these four attitudes and not others. The first consideration from which it is essential to start is that the investigation was conducted during health surveillance— that is, the medical visits that each worker must carry out in the workplace to ensure that he or she is not affected by occupational diseases. The visit interrupts production activity, and this has a cost. The general principle of prevention requires that the costs of prevention should not exceed the benefits. This limits the number of investigations that can be carried out.
Work annoyance, that is, the intolerance that the worker has for certain working conditions, is a recently defined and still very little studied attitude, which, however, is correlated to job stress and the mental health of workers.
Work engagement and overcommitment are the two extremes of the same quantity, which is the amount of skills used in work. Although these two measures come from the same set of resources, they have rarely been investigated simultaneously.
Social capital is a construct dealing with working relationships, taking into account both the individual and organizational dimensions of the company. The binary nature of this variable has led us to prefer it to other measures that investigate relational attitudes, such as social support.
Following the reviewer's advice, we have added this explanation of the choice in the Introduction and added among the Limitations that of not having investigated other attitudes.
We therefore underlined that our study is the first, to our knowledge, to have simultaneously measured the weight of four attitudes on work ability.
In the previous version of the manuscript, we had only mentioned the concept of work ability, referring to the specific literature for further information. However, we realize that it may be useful for the reader to have a greater understanding of the topic, and we have therefore specified it better. Overall, we added 32 references.
MATERIAL AND METHODS
It is unclear whether the instruments used were validated for the population from which the authors collected a sample.
Response: All the tools used had been validated in the local language and had been used in previous surveys carried out in the workplace. We have specified this information in the “Methods” section.
RESULTS
Please indicate the country where the data was collected.
Response: The reviewer is right. We forgot to indicate the country and year in which we collected the observations. We have corrected this omission.
DISCUSSION
This section could be improved if the authors first discussed the results obtained and compared them with previous literature. Then, they could present the study's limitations and suggest avenues for future studies. Finally, they could finish this section by discussing the study's theoretical and practical implications.
R.: Thanks to the reviewer, we have modified the Discussion following the suggested scheme
(x) The English could be improved to more clearly express the research.
R.: The manuscript has been submitted to a native English reader with specific experience in scientific literature. We have added her name in the acknowledgements.
Reviewer 2 Report
Comments and Suggestions for Authors
The manuscript “Individual work attitudes and work ability” is well written and presents a clear, coherent analysis. The methodology is sound, and the results are presented logically and convincingly. The limitations are thoroughly discussed, offering a balanced view of the study's scope and potential weaknesses. The conclusion aligns well with the findings, summarizing the key points effectively.
Below, there are a couple of comments which I suggest the authors to address, in order to improve the manuscript’s overall quality.
- Could the authors specify when the data were collected? This information would be helpful in understanding the context and relevance of the findings (e.g., before or after Covid-19/lockdown).
- The introduction is very brief, and much of the scientific literature is reviewed in the Discussion section. I suggest moving most of it in the introduction section to include a more comprehensive review of relevant literature, which would provide a stronger theoretical foundation for the study.
- Did the authors collect any other data on work characteristics such as job seniority, roles, employment status (full-time vs. part-time), or shift work vs. non-shift work? Including such variables could provide further insights into the factors influencing the study's results.
Author Response
The manuscript “Individual work attitudes and work ability” is well written and presents a clear, coherent analysis. The methodology is sound, and the results are presented logically and convincingly. The limitations are thoroughly discussed, offering a balanced view of the study's scope and potential weaknesses. The conclusion aligns well with the findings, summarizing the key points effectively.
Response: A.: We thank the reviewer for taking the time to review our work and for the appreciation he/she expressed. We tried to take advantage of his/her useful suggestions.
Below, there are a couple of comments which I suggest the authors to address, in order to improve the manuscript’s overall quality.
- Could the authors specify when the data were collected? This information would be helpful in understanding the context and relevance of the findings (e.g., before or after Covid-19/lockdown).
Response: We apologize for forgetting to indicate the country (Italy) and the date of observations (2019, before the pandemic). The information was essential, it has been added.
- The introduction is very brief, and much of the scientific literature is reviewed in the Discussion section. I suggest moving most of it in the introduction section to include a more comprehensive review of relevant literature, which would provide a stronger theoretical foundation for the study.
Response: The reviewer's observation gave us the opportunity to expand the Introduction, better explaining the setting in which the investigation took place, the reasons why we chose some attitudes (and not others), and the importance of work ability. We added 32 references.
- Did the authors collect any other data on work characteristics such as job seniority, roles, employment status (full-time vs. part-time), or shift work vs. non-shift work? Including such variables could provide further insights into the factors influencing the study's results.
Response: As the reviewer correctly guessed, the performance of health surveillance allowed the collection of other occupational and individual data. These were preliminary evaluated but were not included in this article so as not to burden the reader with information that the specific nature of the investigation was not strictly relevant. For example, the performance of night work certainly affects work ability. Night workers (n=306, 23.4% of the sample) have a significantly lower workability score than other workers (7.92+1.80 vs. 8.27+1.73, Mann-Whitney U test p<0.001) but by also adding this variable to the multiple linear regression model the role of overcommitment, vigor and dedication remains the same as reported in Table 3. Similarly, the moderating role of social capital reported in Table 4 remains unchanged. We have added these notes to the text and loaded the tables that also contain night work as a correction factor as supplement S1-S2.
Table S1. Effect of individual attitudes on Work Ability. Hierarchical linear regression models.
|
Model V |
|
Variable |
Beta |
p |
Sex (Female) |
–0.09 |
0.731 |
Age |
–0.055 |
0.041 |
Night work |
-0.060 |
0.021 |
Work Annoyance |
–0.013 |
0.659 |
Social Capital |
–0.006 |
0.843 |
Overcommitment |
–0.086 |
0.002 |
Vigor |
0.226 |
<0.001 |
Dedication |
0.173 |
<0.001 |
Absorption |
0.022 |
0.546 |
Table S2. Moderating effect of Social Capital in the interaction between Work Annoyance and Work Ability.
|
Model III1 |
|
Variables |
Beta |
p |
Work Annoyance |
–0.018 |
0.021 |
Social Capital |
–0.021 |
0.043 |
Annoyance × Social Capital |
0.007 |
0.021 |
Note. 1 Adjusted for age, sex, night work, Overcommitment, Vigor, Dedication and Absorption.
(x) The English is fine and does not require any improvement.
R.: The manuscript has been submitted to a native English reader with specific experience in scientific literature. We have added her name in the acknowledgements.
Reviewer 3 Report
Comments and Suggestions for Authors
I appreciate the opportunity to review the article “Individual work attitudes and work ability”. While it addresses a relevant topic within the field of organizational psychology and occupational health, its structure, methodology, and analysis present numerous deficiencies that compromise its scientific validity. Throughout this review, significant methodological problems, unsupported claims, omission of key aspects in the interpretation of results, and a general lack of critical depth in the analysis of findings have been identified. For this article to be considered for publication, substantial modifications must be made at all levels, from theoretical conceptualization to discussion and presentation of findings.
One of the first issues detected is the lack of precision in the methodological description of the study. In the abstract, it is mentioned that the study investigates the relationship between various work attitudes and work ability, using the Work Ability Score (WAS) and various measurement scales. However, there is no clear specification of the validity and reliability of these tools, nor is their selection justified over other available alternatives in the literature. For example, the measurement of work ability using a single item from the Work Ability Index is questionable, as the full version of the index offers a much more detailed and accurate assessment. The article does not provide solid arguments as to why this abbreviated version was chosen instead of the full questionnaire, nor does it explore potential limitations arising from this decision. Literature has shown that assessing work ability with a single item may be less precise and influenced by contextual and subjective factors, which calls into question the validity of the study’s findings. To improve this section, a clear justification for the choice of instruments should be included, and if abbreviated versions are used, evidence of their equivalence with full versions should be provided.
Another critical problem in the article is the lack of a solid theoretical framework to support the study’s hypothesis. Numerous references are cited without adequate integration into the conceptual development, giving the impression that the introduction is merely an accumulation of previous studies without a coherent argument relating them to the research problem. For example, Hackman and Oldham’s (1976) job characteristics theory is mentioned, but there is no clear connection established between this theory and the studied variables. Concepts are merely stated without explaining how they relate to the study’s theoretical model. Instead of making superficial references to various theories, the introduction should be better structured, presenting a solid argument that justifies the study’s relevance and its contribution to existing literature. In particular, the psychological mechanisms explaining the relationship between work attitudes and work ability should be discussed in more detail, rather than assuming this relationship is self-evident.
The study design also presents major problems. Although it is mentioned that the sample consists of 1,309 workers, insufficient information is provided about the sampling procedure, making it impossible to evaluate the sample’s representativeness. It is unclear whether participant selection was random or if there were inclusion and exclusion criteria that might have introduced biases in the sample composition. This lack of transparency is concerning, as selection bias can significantly affect the external validity of the findings. Additionally, there is no mention of whether a power analysis was conducted to determine whether the sample was large enough to detect significant effects in the employed models. The absence of this information calls into question the robustness of the statistical analyses performed. To improve this section, a detailed description of the sampling procedure should be included, specifying whether it was random, convenience, or stratified sampling, and providing data on response rate and potential non-response biases.
Regarding statistical analysis, the article presents several deficiencies that compromise the validity of the results. It is mentioned that normality tests such as the Kolmogorov-Smirnov and Shapiro-Wilk tests were performed, but it is not justified why parametric models were used instead of robust methods that might be more appropriate given that the data did not follow a normal distribution. This is a serious methodological error, as using parametric techniques on non-normally distributed data can generate biased estimates and affect the interpretation of regression coefficients. Furthermore, the article does not provide confidence intervals for the reported regression coefficients, limiting the evaluation of the precision of the results. There is also no discussion of potential multicollinearity problems among the predictor variables, which is particularly relevant given that several of the analyzed variables could be highly correlated. To improve the robustness of the statistical analysis, a detailed justification of the chosen techniques should be included, confidence intervals for regression coefficients should be reported, and multicollinearity should be verified by calculating the variance inflation factor (VIF).
The study results are presented in a disorganized manner and without adequate interpretation. Multiple correlation tables and regression models are included, but there is no critical analysis of the magnitude of the found effects, nor are they compared with previous studies. Additionally, it is mentioned that social capital moderates the relationship between work annoyance and work ability, but the practical significance of this finding is not explained. It is simply reported that the effect of work annoyance is lower when social capital is high, without exploring the implications of this result for organizational management. To improve this section, results should be contextualized within the existing literature, comparing the magnitude of the found effects with previous studies and discussing their practical relevance in the workplace.
The discussion in the article is particularly weak and lacks critical analysis. Instead of interpreting the results reflectively, the discussion merely repeats the findings without delving into their implications. Alternative explanations for the obtained results are not explored, nor are possible biases that might have influenced the findings discussed. Additionally, key study limitations, such as the fact that the cross-sectional design prevents establishing causal relationships, are not mentioned. The claim that “companies should improve social capital and discourage overcommitment” is overly simplistic and lacks empirical support. To strengthen the discussion, a more detailed analysis of the study’s limitations should be included, alternative explanations for the results should be explored, and future research directions based on the obtained findings should be proposed.
The article’s conclusion is too general and does not provide relevant information beyond what is already stated in the discussion. It is claimed that work attitudes are related to work ability, but there is no clear synthesis of the main findings, nor is their applicability in specific work contexts discussed. Additionally, possible implications for policy-making or organizational strategies based on the study’s results are not mentioned. To improve this section, a concise summary of key findings should be presented, their practical implications should be discussed, and specific recommendations for future research should be proposed.
In conclusion, the article presents numerous methodological, conceptual, and analytical deficiencies that compromise its scientific quality. The lack of a solid justification for the study design, the inappropriate use of statistical techniques, the absence of a well-structured theoretical framework, and the superficial interpretation of results make the manuscript unfit for publication in its current version. A thorough revision is recommended, including a complete restructuring of the introduction, a more rigorous justification of the methodology, a better interpretation of the results, and a more critical analysis in the discussion. Without these improvements, the validity and relevance of the study remain questionable.
Author Response
I appreciate the opportunity to review the article “Individual work attitudes and work ability”. While it addresses a relevant topic within the field of organizational psychology and occupational health, its structure, methodology, and analysis present numerous deficiencies that compromise its scientific validity. Throughout this review, significant methodological problems, unsupported claims, omission of key aspects in the interpretation of results, and a general lack of critical depth in the analysis of findings have been identified. For this article to be considered for publication, substantial modifications must be made at all levels, from theoretical conceptualization to discussion and presentation of findings.
Response: We thank the reviewer for recognizing the importance of the topic we have covered. Unfortunately, his/her request for revision came after EJIHPE had already forwarded the responses of the other two reviewers, who had provided important suggestions useful for improving the manuscript. We had already submitted the updated version, which the reviewer does not know. We have already resolved most of the criticisms the reviewer made regarding our work. However, we took into account the observations he/she sent us and further modified the manuscript in this second revision.
One of the first issues detected is the lack of precision in the methodological description of the study. In the abstract, it is mentioned that the study investigates the relationship between various work attitudes and work ability, using the Work Ability Score (WAS) and various measurement scales. However, there is no clear specification of the validity and reliability of these tools, nor is their selection justified over other available alternatives in the literature.
Response: Accepting the indications of the other reviewers, we explained the reasons for the choice of attitudes, rather than others used in the literature, in the version that we submitted before the reviewer sent his review.
The initial consideration is that the investigation occurred during health surveillance, which refers to the medical examinations that each worker must undergo in the workplace to confirm that they are not suffering from occupational diseases. The visit disrupts production activities, incurring associated costs. The principle of prevention stipulates that the costs associated with preventive measures must not surpass the benefits derived from them. This restricts the quantity of investigations that may be conducted and forces us to select some variables. Furthermore, it forces you to use the short version of the questionnaires.
Work annoyance, defined as the worker's intolerance toward specific working conditions, is a recently identified and under-researched attitude. Nevertheless, it is correlated with job stress and the mental health of employees.
Work engagement and overcommitment represent two extremes of the same construct, specifically the extent of skills utilized in the workplace. Despite originating from the same set of resources, these two measures have seldom been examined concurrently.
Social capital is a concept that addresses working relationships, considering both individual and organizational dimensions within a company. The binary characteristic of this variable has prompted us to favor it over alternative measures that examine relational attitudes, including social support.
Job satisfaction, which some authors include among attitudes, is in our opinion a consequence and not a cause. Therefore, we measured job satisfaction, but we did not analyze this variable.
In accordance with the reviewer's recommendations, we have incorporated an explanation of our choice in the Introduction and noted the limitation of not having explored other attitudes.
In the first version of the article, we had already indicated the reliability of all the instruments used.
This study is, to our knowledge, the first to simultaneously measure the weight of four attitudes on work ability.
For example, the measurement of work ability using a single item from the Work Ability Index is questionable, as the full version of the index offers a much more detailed and accurate assessment. The article does not provide solid arguments as to why this abbreviated version was chosen instead of the full questionnaire, nor does it explore potential limitations arising from this decision. Literature has shown that assessing work ability with a single item may be less precise and influenced by contextual and subjective factors, which calls into question the validity of the study’s findings. To improve this section, a clear justification for the choice of instruments should be included, and if abbreviated versions are used, evidence of their equivalence with full versions should be provided.
Response: In its first version in 2004, the Work Ability Index included a long list of pathological conditions that made this instrument hardly suitable for use in workplaces. A shortened version, which replaced the detail of the pathologies with a list of disease categories, became widespread in the 2010s. The authors' hypothesis that the questionnaire was not unidimensional but could be divided into two or three factors became widely accepted. Thus, the occupational field adopted shortened forms. The shortened form used in this study, the WAS, is the most widely adopted.
The convergent validity between WAS and WAI was statistically significant (rs=0.63) [El Fassi M, Bocquet V, Majery N, Lair ML, Couffignal S, Mairiaux P. Work ability assessment in a worker population: comparison and determinants of Work Ability Index and Work Ability score. BMC Public Health. 2013 Apr 8;13:305. doi: 10.1186/1471-2458-13-305.]. WAI and WAS were both significant predictors of HRQOL and its four dimensions and the explained variance was very similar. The WAI and WAS explained 46% and 44% of the variance related to the HRQOL, respectively. WAI and WAS explained 36-38% and 35-37% of the variance related to dimensions of the HRQOL, respectively. There were significant relationships of both WAS and WAI with job type, work schedule, smoking, and exercise habit [Mokarami H, Cousins R, Kalteh HO. Comparison of the work ability index and the work ability score for predicting health-related quality of life. Int Arch Occup Environ Health. 2022;95(1):213-221. doi: 10.1007/s00420-021-01740-9.]. A heterogeneous development sample (N= 2899) was used to estimate logistic regression coefficients for the complete WAI, a shortened WAI version without the list of diseases, and single-item Work Ability Score (WAS). All three instruments under-predicted the long-term sickness absence LTSA risks in both manual and non-manual workers. The WAI without the list of diseases is a good alternative to the complete WAI to identify non-sicklisted workers. [Schouten LS, Bültmann U, Heymans MW, Joling CI, Twisk JW, Roelen CA. Shortened version of the work ability index to identify workers at risk of long-term sickness absence. Eur J Public Health. 2016;26(2):301-5. doi: 10.1093/eurpub/ckv198.] WAS proved to be the best of the short forms of WAI [Ebener M, Hasselhorn HM. Validation of Short Measures of Work Ability for Research and Employee Surveys. Int J Environ Res Public Health. 2019 Sep 12;16(18):3386. doi: 10.3390/ijerph16183386.]
In the first version of the manuscript, we had already indicated that the high statistical correspondence between WAS and WAI in full form makes this choice valid, common in all epidemiological surveys on large populations.
We indicated among the limitations the fact that the use of a one-item measure may have reduced the power of the associations, which were nevertheless found.
Another critical problem in the article is the lack of a solid theoretical framework to support the study’s hypothesis. Numerous references are cited without adequate integration into the conceptual development, giving the impression that the introduction is merely an accumulation of previous studies without a coherent argument relating them to the research problem. For example, Hackman and Oldham’s (1976) job characteristics theory is mentioned, but there is no clear connection established between this theory and the studied variables. Concepts are merely stated without explaining how they relate to the study’s theoretical model. Instead of making superficial references to various theories, the introduction should be better structured, presenting a solid argument that justifies the study’s relevance and its contribution to existing literature. In particular, the psychological mechanisms explaining the relationship between work attitudes and work ability should be discussed in more detail, rather than assuming this relationship is self-evident.
Response: Responding to the request of the first two reviewers, we had already provided in the first revised version a more complete detail of the meaning of work ability and the fact that numerous factors can contribute to determining the weight of this variable. We also observed that some attitudes could be associated with work ability. We had also explained why we chose some variables rather than others and recalled that there are no studies that simultaneously consider all these variables.
In this revised version we have further detailed the possible relationship between work attitudes and self-assessment of one's work ability.
The study design also presents major problems. Although it is mentioned that the sample consists of 1,309 workers, insufficient information is provided about the sampling procedure, making it impossible to evaluate the sample’s representativeness. It is unclear whether participant selection was random or if there were inclusion and exclusion criteria that might have introduced biases in the sample composition. This lack of transparency is concerning, as selection bias can significantly affect the external validity of the findings. Additionally, there is no mention of whether a power analysis was conducted to determine whether the sample was large enough to detect significant effects in the employed models. The absence of this information calls into question the robustness of the statistical analyses performed. To improve this section, a detailed description of the sampling procedure should be included, specifying whether it was random, convenience, or stratified sampling, and providing data on response rate and potential non-response biases.
Response: In the first version of the manuscript, we indicated that the survey is a census of workers subjected to health surveillance in all the companies we monitor. There are no selection criteria, other than the willingness of the worker to participate or refuse the survey (informed consent). Since the survey is a health promotion activity offered to all workers, there is no evaluation of the sample size. Participation was very high (this data was also in the first version of the manuscript). This supports the possibility of correctly inferring the generality of workers who annually undergo health surveillance in all workplaces.
Regarding statistical analysis, the article presents several deficiencies that compromise the validity of the results. It is mentioned that normality tests such as the Kolmogorov-Smirnov and Shapiro-Wilk tests were performed, but it is not justified why parametric models were used instead of robust methods that might be more appropriate given that the data did not follow a normal distribution. This is a serious methodological error, as using parametric techniques on non-normally distributed data can generate biased estimates and affect the interpretation of regression coefficients.
Response: We run the analyses also using robust methods that allowed us to address non-normality and collinearity issues, and the estimates of the effects relevant to this work were substantially the same. For sake of simplicity, we report in the paper only the results of the ordinary least square (OLS) regressions, while we report in a supplemental material file the comparison between OLS and robust estimates.
Furthermore, the article does not provide confidence intervals for the reported regression coefficients, limiting the evaluation of the precision of the results. There is also no discussion of potential multicollinearity problems among the predictor variables, which is particularly relevant given that several of the analyzed variables could be highly correlated. To improve the robustness of the statistical analysis, a detailed justification of the chosen techniques should be included, confidence intervals for regression coefficients should be reported, and multicollinearity should be verified by calculating the variance inflation factor (VIF).
Response: We added the confidence intervals and effect size measures for each estimated parameter and address multicollinearity as explained in the revised manuscript.
The study results are presented in a disorganized manner and without adequate interpretation. Multiple correlation tables and regression models are included, but there is no critical analysis of the magnitude of the found effects, nor are they compared with previous studies. Additionally, it is mentioned that social capital moderates the relationship between work annoyance and work ability, but the practical significance of this finding is not explained. It is simply reported that the effect of work annoyance is lower when social capital is high, without exploring the implications of this result for organizational management. To improve this section, results should be contextualized within the existing literature, comparing the magnitude of the found effects with previous studies and discussing their practical relevance in the workplace.
Response: We have added effect size measures and better explained the meaning of the interaction effect in the Results section. We then discussed this result in the Discussion section.
The discussion in the article is particularly weak and lacks critical analysis. Instead of interpreting the results reflectively, the discussion merely repeats the findings without delving into their implications. Alternative explanations for the obtained results are not explored, nor are possible biases that might have influenced the findings discussed. Additionally, key study limitations, such as the fact that the cross-sectional design prevents establishing causal relationships, are not mentioned. The claim that “companies should improve social capital and discourage overcommitment” is overly simplistic and lacks empirical support. To strengthen the discussion, a more detailed analysis of the study’s limitations should be included, alternative explanations for the results should be explored, and future research directions based on the obtained findings should be proposed.
Response: In the first draft of the article, we made it clear that one of the limitations was that the cross-sectional nature of the data meant that we couldn't prove causality. However, we also stressed that aptitudes are natural traits that come before work ability and that the statistical method we used (hierarchical regression) let us show how important the associations are.
Since the study presented here shows that social capital is positively associated with work ability while overcommitment is negatively associated, and the literature confirms these associations, we believe this is sufficient support for health promotion programs. We do not understand why the reviewer, or the AI program he/she used, claims otherwise.
Accepting the reviewer's invitation, we have expanded the section on limitations and the one on prospects for the study.
The article’s conclusion is too general and does not provide relevant information beyond what is already stated in the discussion. It is claimed that work attitudes are related to work ability, but there is no clear synthesis of the main findings, nor is their applicability in specific work contexts discussed. Additionally, possible implications for policy-making or organizational strategies based on the study’s results are not mentioned. To improve this section, a concise summary of key findings should be presented, their practical implications should be discussed, and specific recommendations for future research should be proposed.
Response: we added a summary of the results at the beginning of the Discussion section and tried to make more explicit the implications for policy making.
In conclusion, the article presents numerous methodological, conceptual, and analytical deficiencies that compromise its scientific quality. The lack of a solid justification for the study design, the inappropriate use of statistical techniques, the absence of a well-structured theoretical framework, and the superficial interpretation of results make the manuscript unfit for publication in its current version. A thorough revision is recommended, including a complete restructuring of the introduction, a more rigorous justification of the methodology, a better interpretation of the results, and a more critical analysis in the discussion. Without these improvements, the validity and relevance of the study remain questionable.
Response: We have expanded the Conclusions section

Round 2
Reviewer 1 Report
Comments and Suggestions for Authors
Overall, the authors have addressed my previous comments. As such, I have nothing more substantial to add. The only exception concerns the introduction section; I would recommend adding more clearly the definitions of each studied variable, as they did in the author's response to my report. In other words, add more clearly these definitions:
Work annoyance, that is, the intolerance a worker feels towards certain working conditions, is a recently defined and still poorly studied attitude correlated with job stress and workers' mental health.
Work engagement and overcommitment represent two extremes of the same continuum, which encompasses the number of skills used at work. Although these two measures derive from the same set of resources, they have seldom been explored simultaneously.
Social capital is a construct that pertains to working relationships, considering a company's individual and organisational dimensions. Because it is binary, we favour it over other measures that examine relational attitudes, such as social support.
Moreover, concerning social capital, explain this binary nature better in the introduction section. In addition to all the definitions introduced in the introduction section, add the references used.
Author Response
Overall, the authors have addressed my previous comments. As such, I have nothing more substantial to add. The only exception concerns the introduction section; I would recommend adding more clearly the definitions of each studied variable, as they did in the author's response to my report. In other words, add more clearly these definitions:
Work annoyance, that is, the intolerance a worker feels towards certain working conditions, is a recently defined and still poorly studied attitude correlated with job stress and workers' mental health.
Work engagement and overcommitment represent two extremes of the same continuum, which encompasses the number of skills used at work. Although these two measures derive from the same set of resources, they have seldom been explored simultaneously.
Social capital is a construct that pertains to working relationships, considering a company's individual and organisational dimensions. Because it is binary, we favour it over other measures that examine relational attitudes, such as social support.
Moreover, concerning social capital, explain this binary nature better in the introduction section. In addition to all the definitions introduced in the introduction section, add the references used.
Response: We thank the reviewer for the attention with which he/she has sought the improvement of the article since the first submission. We have added the following specifications:
Work annoyance: the level of irritation or frustration experienced by workers regarding specific aspects of their job. It pertains to the working conditions, namely, factors such as the physical work environment, safety, and available resources that can influence a worker's comfort and efficiency, and to the cognitive demands of the work, namely, the mental requirements of the job, including workload, complexity of tasks, and the need for continuous learning (Magnavita & Chiorri, 2022).
Work engagement is defined as a positive, fulfilling, work-related state of mind characterized by vigor, dedication, and absorption (e.g., Demerouti et al., 2002). Vigor refers to high levels of energy and mental resilience while working, the willingness to invest effort in one's work, and persistence in the face of difficulties. Dedication comprises a sense of purpose, passion, motivation, pride, and challenge. Absorption is the state of being fully absorbed and enjoying one's task, which makes time fly by and makes it difficult to break away from it.
Social capital refers to a multifaceted psychological construct that emphasizes the value of interpersonal relationships and the resources, advantages, and opportunities that people can access through their social networks, relationships, and social interactions at work. It can be thought of as a dynamic psychological resource that, through reciprocity, trust, shared standards, and social networks, promotes both individual and community results. This term refers to the resources a person may be able to access through their social connections. These resources are determined by the structure and pattern of social network links (structural dimension), the state of interpersonal connections, which includes emotional ties, trust, and respect (relational dimension), and the shared understanding, interpretations, and meanings among social groups (cognitive dimension). (Ehasn et al., 2019; Nolzen, 2018).
Reviewer 3 Report
Comments and Suggestions for Authors.
Author Response
Comments and Suggestions for Authors
.
Response: The reviewer gave no suggestions, he/she only put a point. In the first round he/she had used artificial intelligence. This behavior is not in line with what EJIHPE requires from reviewers. The reviewer's lack of cooperation prevents us from responding to his/her criticisms.